# Nickel Porous Compacts Obtained by Medium-Frequency Electrical Resistance Sintering

**DOI:** 10.3390/ma13092131

**Published:** 2020-05-04

**Authors:** Fátima Ternero, Eduardo S. Caballero, Raquel Astacio, Jesús Cintas, Juan M. Montes

**Affiliations:** Engineering of Advanced Materials Group, Higher Technical School of Engineering, University of Seville, Camino de los Descubrimientos, s/n, 41092 Sevilla, Spain; esanchez3@us.es (E.S.C.); rastacio@us.es (R.A.); jcintas@us.es (J.C.); jmontes@us.es (J.M.M.)

**Keywords:** electrical resistance sintering, electrical consolidation, MF-ERS, ECAS, FAST, hot pressing, sintering, nickel powder, powder metallurgy

## Abstract

A commercially pure (c.p.) nickel powder was consolidated by Medium-Frequency Electrical Resistance Sintering (MF-ERS). In this consolidation technique, a pressure and the heat released by a high-intensity and low-voltage electrical current are concurrently applied to a metal powder mass. A nickel powder with a high tap porosity (86%) and a low applied pressure (only 100 MPa) is chosen in order to be able to obtain compacts with different levels of porosity, to facilitate the study of the porosity influence on the compact properties. The influence of current intensity and heating time on the global porosity values, the porosity and microhardness distribution, and the electrical conductivity of the sintered compacts is studied. The properties of the compacts consolidated by MF-ERS are compared with the results obtained by the conventional powder metallurgy route, consisting of cold pressing and furnace sintering. A universal equation to describe the porosity influence on all the analyzed properties of powder aggregates and sintered compacts is proposed and validated.

## 1. Introduction

Probably, the first patent on the electrical consolidation (no pressure applied) of a powder mass dates from 1906 [1]. Since then many other variants have been developed, all with the aim, explicit or implicit, of using them on an industrial scale (see Grasso et al. [2,3], Orrù et al. [4], and Olevsky et al. [5]). Perhaps the most popular technique nowadays is the so-called Spark Plasma Sintering (SPS), in which graphite (electrically conductive and low wear resistance) dies and punches are used, a combination of AC and DC current is applied, and a controlled vacuum atmosphere is required. This does not result in cheap and attractive equipment for industry. To make the process more cost-effective, more durable alumina dies (electrically insulating) could be used instead, as well as cheaper equipment by using minimally adapted industrial resistance welding equipment. In fact, the modality known as Electrical Resistance Sintering (ERS), developed by Taylor [6] and widely studied by Lenel [7], takes advantage of these elements.

Comparing the ERS technique with the conventional powder metallurgy (P/M) route of cold-pressing and furnace sintering, three aspects are noteworthy: (i) the high densification rates achieved with the ERS at low pressures (around 100 MPa), (ii) the very short processing times (around 1–2 s), and (iii) the possibility of not using protective atmospheres, as a consequence of (ii). However, the usual non-homogeneous temperature distribution, inside the compacts, during ERS (or a similar technique) makes it difficult to achieve a homogeneous microstructure (and isotropic properties) throughout the compact [8,9,10,11]. In addition, finding a suitable material that provides acceptable cost and durability for the dies is also a problem [12].

In this work, an equipment of Medium-Frequency Electrical Resistance Sintering (MF-ERS) obtained by properly modifying a resistance welding machine has been employed. The medium-frequency technology offers a great advantage, as it allows using a DC current and smaller and lighter transformer cores, without power loss. Although this technique has not yet enjoyed a great deal of theoretical development by many researchers, some complete theoretical models and simulations of the process can be found in [13,14].

Nickel was chosen for its potential future applications [15]. Nowadays, a large amount of pure nickel powder is produced to obtain alloys by powder metallurgy as it improves some mechanical properties such as ductility and tensile strength. Nickel-based superalloys are also used for applications requiring high corrosion resistance, high strength, and good toughness over wide temperature ranges. These superalloys can also be produced by powder metallurgy. This material has already been studied in the context of electric sintering techniques in [7,16,17]. This work could provide a first approach to know the pros and cons of applying MF-ERS technique in the manufacturing of parts from high nickel content powders.

Porosity and microhardness distribution and electrical resistivity of all the electrically consolidated compacts were determined and compared with those obtained from compact prepared through the conventional P/M route of cold pressing and furnace sintering.

As a secondary aim, this work also constitutes a general validation that the influence of porosity (i.e., the void volume fraction) on all analyzed properties of sintered compacts can be expressed through the law:(1)$$p=p_{0}{\left(1-\Theta /\Theta_{M}\right)}^{n}$$
where *p* is the property of the material with porosity Θ, *p*_0_ is the value of this property for the fully dense material, Θ_M_ is the tap porosity of the powder with which the compact was manufactured, and *n* is a fitting parameter. Note that, considering *n* is a positive number, Equation (1) satisfies the expected boundary conditions, *p* → *p*_0_ as Θ → 0, and *p* → 0 as Θ → Θ_M_, since, in this last situation, interparticle contacts are points. If *n* was a negative number, then *p* → *p*_0_ as Θ → 0, and *p* → ∞ as Θ → Θ_M_, for the same reason. Equation (1) has been previously proposed in various contexts and by different authors [18,19,20].

## 2. Experimental Procedure and Materials

### 2.1. MF-ERS Equipment

To carry out the electrical consolidation experiments by MF-ERS, a properly adapted press type resistance welding machine (Serra Soldadura S.A., Barcelona, Spain) was employed. This equipment incorporates a three-phase 1000 Hz and 100 kVA transformer, and control electronics capable of providing a current output of a previously set value, and a servo-driven upper head capable of producing a maximum load of 15 kN. The equipment is also conveniently instrumented to access the evolution of the relevant process parameters. Thus, the values of the current intensity, the voltage between the equipment plates, and the position of the upper head are monitored and recorded during the MF-ERS experiments. 

In addition to the adapted welding machine (the power and pressure source), the electrical consolidation process also requires a die containing the powders to be sintered, and the punches/electrodes to apply pressure and electrical current (Figure 1). Following the design employed by Lenel [7], a die consisting of an alumina tube (12 mm inner diameter), externally reinforced by a steel ring (hoop), was used. Each punch/electrode consisted of a disk (wafer) of heavy metal tungsten alloy (24.6 wt % Cu–75.3 wt % W), with good non-stick and electro-erosion resistance properties, and a cylindrical bar made of a temperature-resistant Cu-alloy (0.1 wt % Zr, 1 wt % Cr and 98.9 wt % Cu). The wafers also have lower thermal conductivity than the cylindrical bars, thus slowing down the leakage of heat generated in the compact to the water-cooled bedplates.

### 2.2. Experimental Procedure

#### 2.2.1. Powder Characterization

Apparent and tap density were determined according to MPIF Standards [21,22]. Morphometric and granulometric aspects of the selected powder were determined with the help of high-resolution SEM micrographs (FEI Teneo, FEI Company, Hillsboro, OR, USA). The granulometry was also carried out with the help of laser diffraction (Mastersizer 2000, Malvern Panalytical Ltd., Malvern, UK).

The powder metallurgy characteristics of the powder were completed with the measurement of its compressibility [23], and the variation of its electrical conductivity vs. porosity. For this last determination, the device shown in Figure 2 was used, according to the procedure described in [24].

#### 2.2.2. MF-ERS Process

In order to reduce friction between wall die and powder mass, a suspension of graphite in acetone was deposited on the inner die wall as a very thin layer, acting both as lubricant and non-sticking agent. Before each electrical consolidation, the die was shaken in order for the contained powder to reach its tap porosity.

A MF-ERS consolidation process starts with a cold pressing stage for 1000 ms. In this time, a constant pressure (100 MPa, in this work) is applied to the powder mass, but no current is passing through. The next step is a heating stage, where, in addition to pressure (100 MPa), a current intensity is applied. The last step consists of a cooling stage for 300 ms, when again only pressure (100 MPa) is applied. The whole process is carried out in the air, without atmosphere control.

Electrical consolidation experiments were carried out with heating times of 400, 700 and 1000 ms and current intensities of 6, 8 and 10 kA. These intensities, normalized with the compact cross-section, represent current densities of 5.31, 7.07 and 8.84 kA/cm^2^. These processing conditions were sufficient to achieve high densities in MF-ERS experiments with iron powder [8]. The 3 g of powder mass used in the experiments made the compact reach a height/diameter aspect ratio close to 1/2.

Also, compacts of 3 g mass and 12 mm diameter were conventionally consolidated for comparison purpose. Firstly, they were cold compacted at 1200 MPa and then they were vacuum (5 Pa) furnace sintered at 800 ºC for 30 min. The properties of the conventional compacts will serve as references for the study of the porosity influence, at the limit of low porosities.

#### 2.2.3. Compacts Characterisation

The MF-ERS compacts final porosity was calculated from the final sizes and weight of the specimens. The porosity values determined by this method have an uncertainty of approximately 5%. On the other hand, the diametral section of all the compacts (electrically and conventionally consolidated) was analyzed by optical microscopy (EPIPHOT 200, Nikon, Tokyo, Japan) in order to study the porosity distribution. The non-homogeneous porosity distribution is a consequence of the temperature achieved in different compact zones during the process [8,14].

The Vickers microhardness was measured with a microhardmeter (DURAMIN-A300, Struers GmbH, Willich, Germany) using a load of 1 kg and according to [25]. The Vickers microhardness of compacts was measured on a diametral section quadrant in five different points, as shown in Figure 3 (different measurements are needed because of the non-uniform porosity distribution). Other quadrants are supposed to behave in a similar way due to the symmetry of the compact. 

In addition, the electrical resistivity on the compact base was determined. A four points probe and a Kelvin bridge (Micro-ohmmeter, CA 6240, Chauvin Arnoux, Paris, France) were used for the electrical resistance measurements at room temperature, with a measuring range of 0.01 μΩ–1 Ω (Figure 4). The thermoelectric effects were cancelled by changing the probes polarity and obtaining the mean value of two measures for each specimen.

For the probe electrodes spacing (*d* = 2 mm), the electrical conductivity can be calculated as [26]:(2)$$\sigma ={\left(2\pi dR\right)}^{-1}$$

The relative error in the conductivity computation, according to the uncertainty in the measurement of the resistance values, is always lower than 7%.

### 2.3. Material

A commercial pure nickel powder obtained by carbonyl refining and marketed by the Valve INCO under the name Ni Type 255 was selected for this work. The main impurities are 0.001 wt % S, 0.01 wt % Fe, 0.15 wt % O, and 0.3 wt % C. A low apparent density of only 0.6 g/cm^3^ (6.7% of the absolute density, 8.91 g/cm^3^) and a tap density of 1.25 g/cm^3^ were measured. This last value results in a tap porosity of 0.86. Nickel powder Ni Type 255 has a structure formed by fine three-dimensional filaments, similar to a necklace. This structure can be seen in the high-resolution SEM images shown in Figure 5.

From these micrographs it has been possible to determine that the particle size (bead of the necklace) of nickel Ni Type 255 powder is 3–5 μm. The filamentous morphology favors the agglomeration of the powder in small groups up to 50 μm in size. The mean size of these agglomerates obtained by particle size diffraction technique was d(4,3) = 23 μm.

As an additional characterization of the powder, its compressibility curve (Figure 6a) and its conductivity vs. porosity curve (Figure 6b) were determined. 

As can be seen, the theoretical curves in Figure 6 fit reasonably well with the experimental curves (with *R*^2^ coefficients of 0.9977 for pressure, and 0.9998 for electrical conductivity). These theoretical curves have been obtained by fitting, by means of least squares method, the experimental data points with equations of type Equation (1).

Thus, the relationship between effective pressure (*P*) and porosity (Θ) is as follows:(3)$$P=P_{0}{\left(1-\Theta /\Theta_{M}\right)}^{n}$$
where Θ_M_ = 0.869, *P*_0_ = 3113.116 and *n* = 3.7524. (The *P*_0_ value represents the pressure required to achieve zero porosity). For the applied pressure of 100 MPa, the porosity of the powder mass was 0.55 (indicated in Figure 6a).

On the other hand, for the electrical conductivity:(4)$$\sigma =\sigma_{0}{\left(1-\Theta /\Theta_{M}\right)}^{m}$$
where Θ_M_ = 0.869, *σ*_0_ = 1.582 × 10^6^ and *m* = 1.909. (Note that the value of *σ*_0_ is lower than that of the corresponding value of pure nickel [27], about 1.43 × 10^7^ (Ω·m)^−1^, which means that *σ*_0_ also takes into account the effect of oxide layers that get in the way of contact between particles). For the applied pressure of 100 MPa, the measured value of the electrical conductivity was 5.201 × 10^5^ (Ω·m)^−1^.

## 3. Results and Discussion

### 3.1. Final Porosity and Specific Thermal Energy

After the application of 100 MPa, the porosity of the powder mass was reduced from 0.86 (the tap porosity value) to 0.55. After the passage of the electrical current, the porosity was further reduced. The final porosity Θ_F_ of the MF-ERS compacts is shown in Table 1, as a function of the current intensity (*I*) and the heating time (*t_H_*). 

Porosities in Table 1 follow an expected trend, decreasing downwards and to the right of the Table. For a better analysis, data in Table 1 have been represented in Figure 7.

Figure 7 shows clear linear trends, parallel to the global trend, for both current intensity and heating time families. From the comparison of the two graphs, in the tested ranges, the current intensity has a greater effect than the heating times on the final porosity of the compacts.

Naturally, it is to be expected that there is a correlation between the thermal energy released in each experiment and the final porosity achieved. The Joule thermal energy released per powder unit mass, which will be called the *specific thermal energy* (*STE*), can be computed by integrating the dissipated electrical power during the heating time. That is,
(5)$$STE=\frac{1}{M}\int_{0}^{t_{H}}I(t)\cdot V(t)dt$$
where *M* is the powder mass, *I* the current intensity passing through the powder, and *V* is the electrical voltage drop through the powder column. 

*STE* values are shown in Table 2, following the expected behaviour, with greater values for higher intensities and heating times.

A graph of the final porosity (Θ_F_) vs. *STE* is shown in Figure 8. The trend is, again, predictable, obtaining lower porosities for higher *STE* values.

The trend is clearer by studying the obtained data grouped by families of intensities and heating times, following marked trends. On the one hand, the icons with the same color (representing families with identical heating times) follow trends parallel to the general trend of the whole scatter plot (continuous line), which means that, within each family, *STE* values grow with intensity. These trends are, moreover, stratified in increasing order (400, 700 and 1000 ms), this last series being the one that implies higher *STE* values.

On the other hand, icons with the same geometric shapes in the graph are also correlated, which means that families with identical intensity follow parallel trends. It happens that higher porosities are found for 6 kA, as a result of the release of less thermal energy, and lower porosities are found for 10 kA, as a result of the release of greater thermal energy. However, a similar porosity of about 0.36 can be reached with 10 kA-400 ms or 8 kA-1000 ms, although with the second combination, a 56% extra *STE* is necessary. 

In order to complete the information, it would be interesting to provide, at least, the values of the mean temperatures inside the compact. Unfortunately, due to the nature of the MF-ERS technology, it is not possible to reliably measure the temperature inside the compact during the process. Only by means of theoretical estimation is it possible to know the achieved temperatures. Using simulators developed in previous works [13,14], it has been possible to establish that the maximum values of the mean temperatures reached, for the analyzed conditions, move in the range of 400 to 700 K. These values may seem low, but it must be considered that the local temperature may be much higher in the interparticle contact areas. Unfortunately, there are no reliable theoretical estimates for these local temperatures.

The final porosity of the conventional compacts was about 9%, a much lower value than that achieved by the MF-ERS compacts. It should be noted that, in the latter case, the pressure was 10 times lower. However, it can be said that with these same processing conditions, but using iron powder, it was able to obtain final porosities of 6%. The apparent ineffectiveness of the MF-ERS process with this nickel powder is related to its high tap porosity. In order to obtain much lower porosities, more severe processing conditions should be adopted: higher intensities, and/or longer times, and/or higher applied pressure.

### 3.2. Porosity Distribution

The peculiarities of the MF-ERS process result in micro and macro-structural characteristics very different from those observed in conventionally P/M produced compacts. Figure 9 shows the diametral sections of a conventional press and furnace sintered compact, and an electrically consolidated compact (8 kA-400 ms). The conventional compact shows a relatively uniform porosity of approximately 9%, with a narrower, more porous edge on the periphery. The compact consolidated by MF-ERS reveals a greater porosity with non-uniform distribution, because of the heterogeneous temperature distribution, which is higher in the center of the compact. As expected, both the wafers and electrodes (in contact with the cooling plates) and the die walls act as heat sinks, and the lower the temperature reached, the higher the porosity.

To reduce the final porosity of the compact, the current intensity and/or the heating time must be increased, and, in principle, the increase of the applied pressure can also help. However, increasing the applied pressure does not always achieve the desired effect. A higher pressure leads to a higher density in the first moments of the heating period. This reduction can be counterproductive as it will lead to a decrease in the electrical resistivity of the powder mass, which will result in less thermal energy being released, and therefore less softening of the material and higher final porosity.

High pressure values can cause another undesirable effect that will also lead to higher final porosity. When the process is carried out with adequate pressure, the center of the compact, at higher temperature, collapses first. The gas contained in the pores of this zone (or even that which may evolve due to the heating) can be evacuated through the (open) porosity of the peripheral zones, which are kept at a lower temperature. Thus, in the correct operation of the MF-ERS, porosity is progressively removed from the interior towards the outer. The peripheral porous zones will become narrower and the central area denser, increasing uniformity over time. On the other hand, when the pressure value is too high, the porosity of the peripheral zones may close, preventing the subsequent evacuation of the gases inside the compact. The gas that is occluded at a very high temperature can generate enough counter-pressure to prevent the densification of the compact. Therefore, the choice of pressure can be critical and it cannot be said that increasing it will result in a guaranteed decrease in the final porosity. Therefore, if the aim was to reduce the final porosity of the samples (which was not the case in this paper), intensities and times would have to be increased. Obviously, if conditions are too severe, the compact core could become molten [13].

Figure 10 shows a set of micrographs taken at the upper left corner of the diametral cross-section of the MF-ERS compacts. Figure 11 shows a set of micrographs taken at the center of the compacts. The presence of a more porous periphery can be observed, and it can also be seen that the porosity decreases, in general, as the current intensities and time increase. However, for the intensity of 6 kA, the irregularities are abundant and may cause fracture due to poor consolidation.

It can be noticed (in Figure 10 and Figure 11) that the pores in the central area generally have a more rounded appearance than those in the peripheral areas. Naturally, this fact becomes more evident as the processing conditions (intensity and heating time) become more severe, because of the higher temperatures reached.

It is not easy to predict, for certain processing conditions, the final porosity that the compacts will reach and even less the porosity distribution. The process of the hot densification that takes place during MF-ERS is coupled with the simultaneous release and transfer of heat. In addition to the processing conditions, the number of interconnected factors (powder composition and morphometric aspects, die and punches materials, compact aspect ratio, etc.) is so great that the only way to advance is by means of numerical simulations. Only with these ones, is it possible to predict the temperature and porosity distribution in the system. This will be the work to be carried out, specifically for the tested conditions of this paper, in the near future.

### 3.3. Microhardness Distribution

Microhardness of conventional processed compacts resulted in a mean value of 97 HV1, and a standard deviation of 5 HV1. The mean microhardness values, resulting from the five measurements of each specimen, for the different MF-ERS compacts, are shown in Table 3. It can be observed that the mean microhardness increases with the current intensity and the heating time.

Figure 12 shows the microhardness in relation to the final porosity of the compacts. It can be clearly seen how the mean microhardness increases with the reduction in porosity. The continuous red line has been drawn fitting Equation (1) to the data points. The result of fitting by least-squares method is:(6)$$HV=HV_{0}{\left(1-\Theta /\Theta_{M}\right)}^{n}$$
where Θ_M_ = 0.869, *HV*_0_ = 401.44 and *n* = 3.442 with *R*^2^ = 0.90. (The *HV*_0_ represents the *HV* value for the full density material).

Comparing conventional and electrical consolidation, it can be deduced that the former can provide higher microhardness values. However, the projection of the trend of microhardness of electrically consolidated compacts (see Figure 12) for the same value of porosity (9%) would give a much higher value than that shown by the conventional compact (264 versus 97.1). This may indicate that the microstructure changes during the high temperature heat treatment of conventional sintering. In this case, exposure to high temperature is sufficiently long, and the so small initial size of powder, that significant grain growth can be triggered. This would be associated with a considerable decrease in hardness. The MF-ERS process would not show this phenomenon due to its comparatively short duration. It should also not be ruled out that the rapid cooling provided by the MF-ERS technique, in which the metallic electrodes are water-cooled, causes in the compact a rapid contraction and high concentration of dislocations and stresses, resulting in higher hardness.

### 3.4. Electrical Resistivity

The conductivity of the bulk pure nickel is 1.43 × 10^7^ (Ω·m)^−1^, according to [27], whereas that of the conventionally consolidated compact was slightly less, 1.12 × 10^7^ (Ω·m)^−1^. The difference can be mainly attributed to the presence of porosity in the conventional press and sintered specimen. In addition, a deficient sintering produces non perfect metal–metal contacts between particles, which is due to the oxide layers surrounding particles. (Another source of minor discrepancy can be due to presence of impurities in the bulk.) The presence of oxide nanometric films on the surface of metallic parts is a well-known fact in P/M and in the Metallic Corrosion field [28,29,30]. In P/M, these oxide layers covering the powder particles are considered as drawbacks that hinder (and sometimes prevent) the sintering process [31,32].

Table 4 gathers the conductivity values of the MF-ERS compacts. As expected, because of the decrease in porosity and better particles bonding, the electrical conductivity increases by increasing the current intensity and/or the heating time. (For comparison purposes, the electrical conductivity of conventionally consolidated compact and green compact (without sintering) are 1.12 × 10^7^ (Ω·m)^−1^ and 2.36 × 10^5^ (Ω·m)^−1^, respectively).

The electrical conductivity as a function of porosity can be adjusted by means of least squares with Equation (4). The resulting parameters of this fitting are Θ_M_ = 0.869, *σ*_0_ = 1.061 × 10^7^ and *m* = 1.937, with *R*^2^ = 0.922.

Figure 13 shows the variation of the electrical conductivity with porosity. To facilitate the analysis, three curves obtained with Equation (4) have been represented, but with different values of its parameters: (i) the green dotted curve represents the variation of the conductivity of the loose powder subjected to compression; (ii) the red solid curve represents the conductivity as a function of the porosity of an aggregate of bare particles (without surface oxides layers), which according to previous studies [18,19] can be obtained assuming the values *σ*_0_ = *σ*_M_ (the bulk conductivity) and *n* = 3/2; and (iii) the red dotted curve represents the variation, with the porosity, of the MF-ERS compacts conductivity. The point representing the conventionally obtained compact is also added.

As can be seen, Figure 13 gives a good idea of how the electrical sintering process works. It can remove the oxide layers, although not as effectively as the conventional process, in which the metal–metal contacts between particles practically resembles what a compact of (deoxidized) bare powder would have.

## 4. Conclusions

In this work, a commercially pure nickel powder with high tap porosity (86%) has been successfully consolidated using the MF-ERS technique. Within the analyzed conditions window (varying current intensity and heating time, for an applied pressure set at 100 MPa), the porosity of the compacts obtained varies between 45% and 32%, which makes this technique an excellent way of obtaining porous and perfectly cohesive compacts. 

This concludes that the non-uniform distribution of temperature inherent to MF-ERS processing produces a heterogeneous distribution of porosity. This work allows to understand the important role played in the MF-ERS technique by the (dielectric) oxide layers that surround the metal powder particles. The work also allows to conclude that more severe conditions are required (higher intensities and/or higher times and/or higher applied pressures) to achieve much more densified compacts from such a high impact porosity powder (0.86). 

The work also allows us to conclude the suitability of the equation proposed for the description of the influence of porosity (Equation (1)) on all the studied properties, as well as the important role played by the tap porosity value in this description.

## Figures and Tables

**Figure 1 materials-13-02131-f001:**
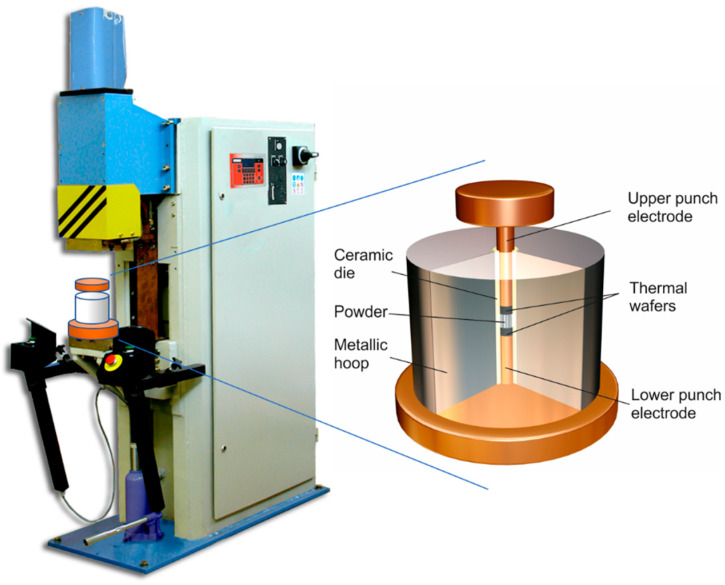
Electrical resistance welding machine adapted to act as the MF-ERS equipment, and sketch of the die and electrodes set employed in the experiments.

**Figure 2 materials-13-02131-f002:**
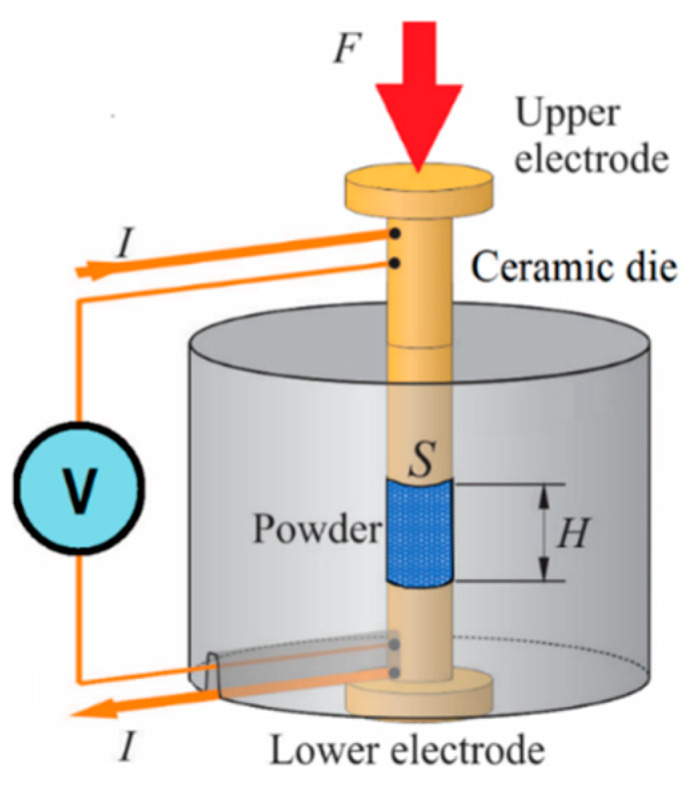
Scheme of experimental device used to determine the conductivity vs. porosity curve of a powder mass under pressure.

**Figure 3 materials-13-02131-f003:**
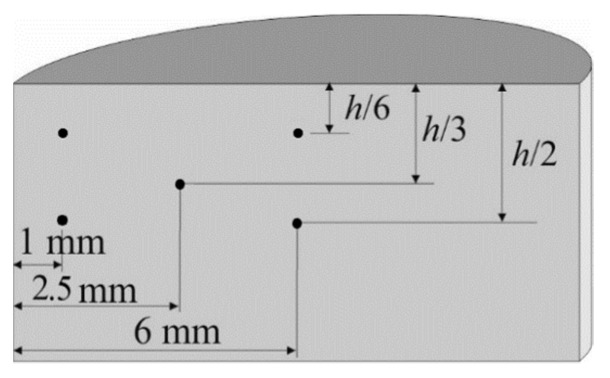
Microhardness indentation map on a compact diametral section. All the measured values were averaged to determine the mean microhardness.

**Figure 4 materials-13-02131-f004:**
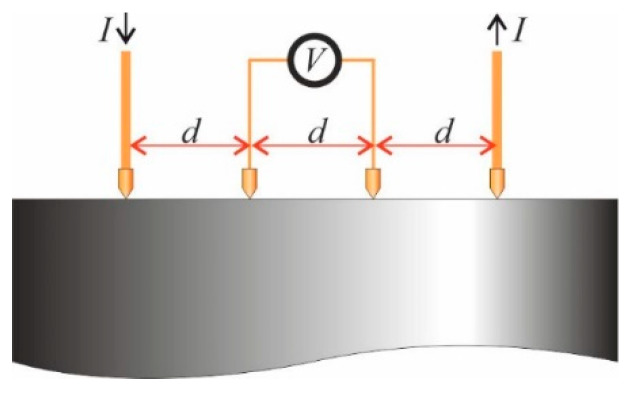
Scheme of the four-point probes used for the determination of the electrical resistivity. The four electrodes are connected to a Kelvin bridge that supplies the ratio *V*/*I* (that is, *R*) from which the resistivity can be calculated.

**Figure 5 materials-13-02131-f005:**
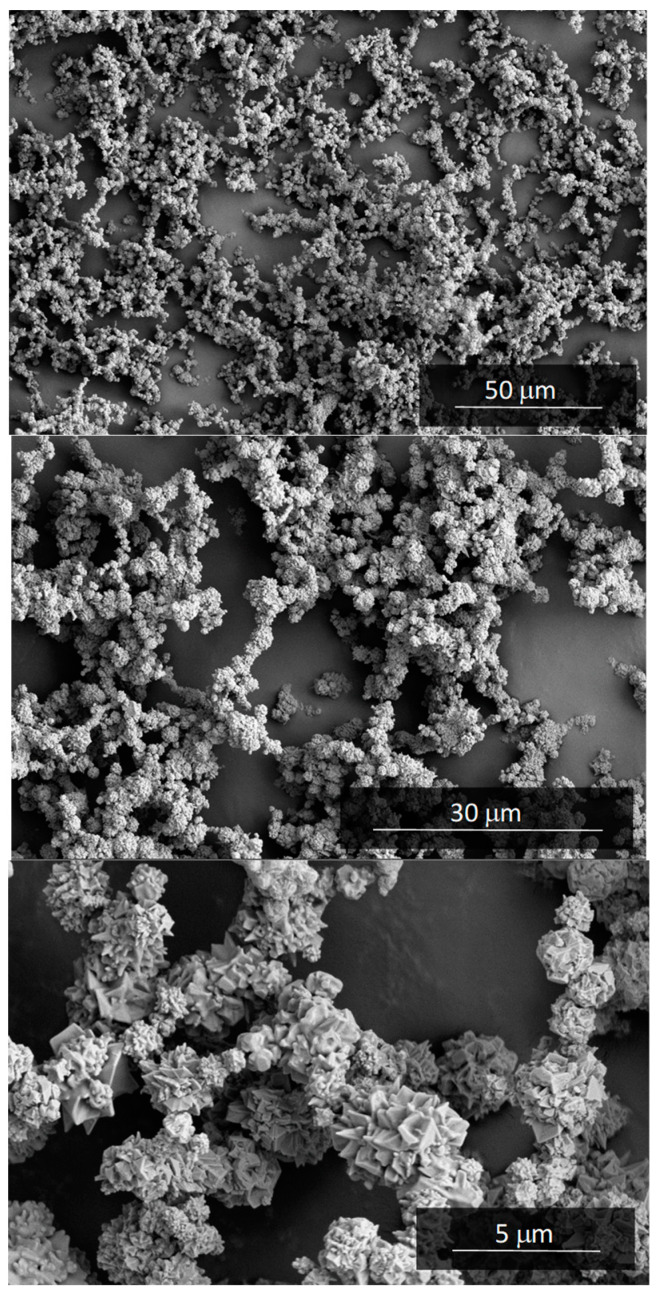
SEM micrographs (Fei Teneo) of the Ni Type 255 powder used for MF-ERS experiments.

**Figure 6 materials-13-02131-f006:**
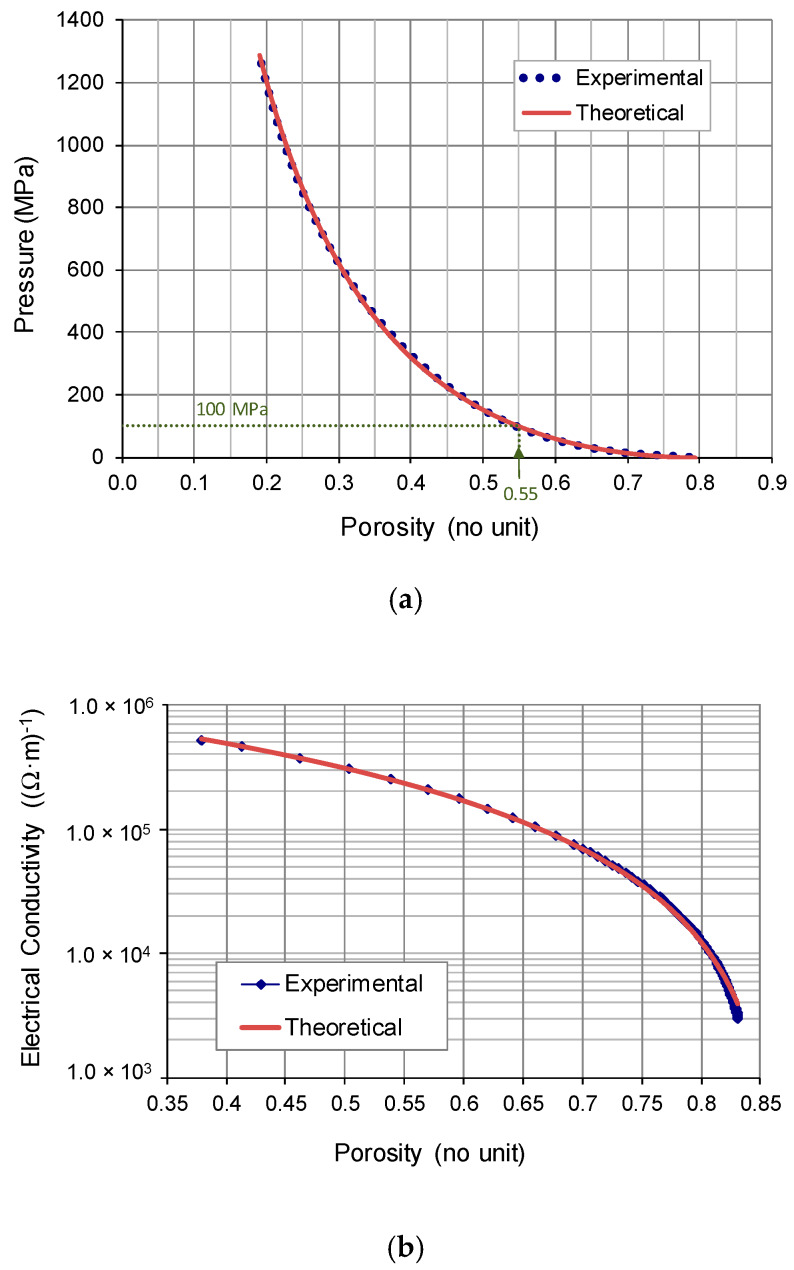
(**a**) Compressibility curve, and (**b**) Electrical conductivity vs. porosity, for a Ni Type 255 powder mass.

**Figure 7 materials-13-02131-f007:**
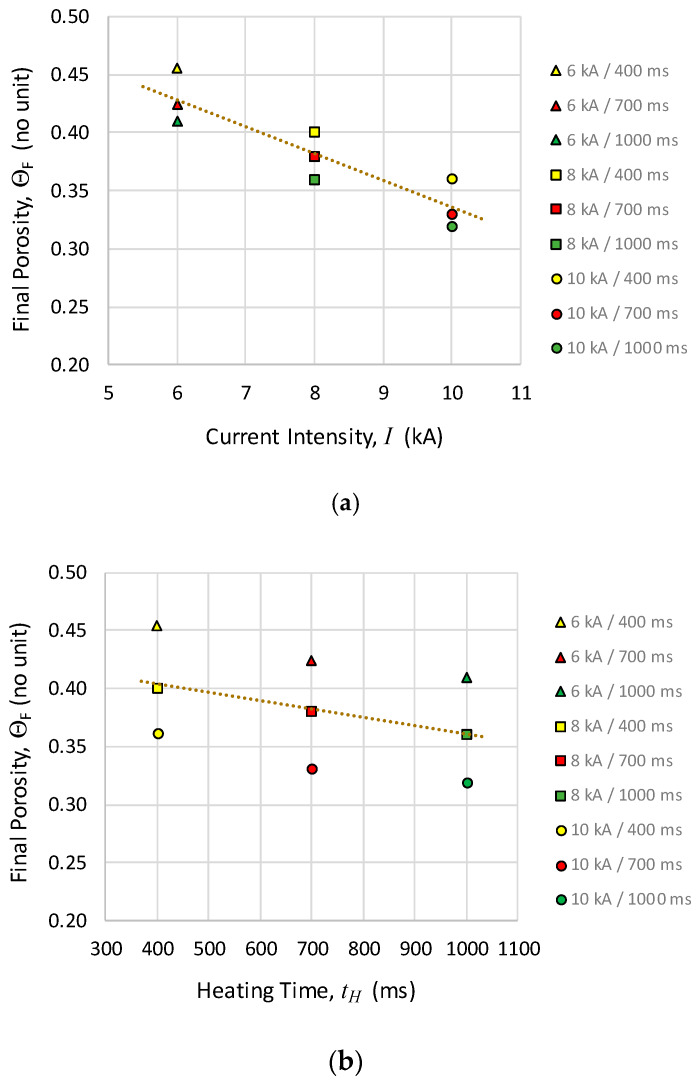
Final porosity of the compacts (Θ_F_) against: (**a**) the current intensity and (**b**) the heating time, for the different MF-ERS experiments. The dotted lines in both graphs represent the trend of the whole set of points.

**Figure 8 materials-13-02131-f008:**
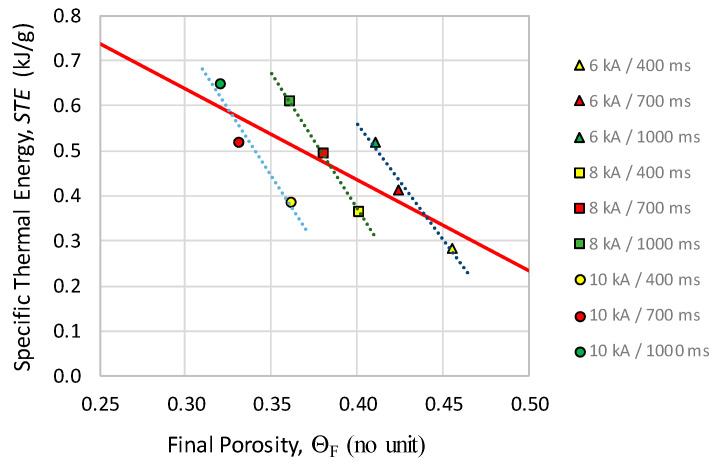
Final porosity (Θ_F_) versus Specific Thermal Energy (*STE*) for the different MF-ERS compacts.

**Figure 9 materials-13-02131-f009:**
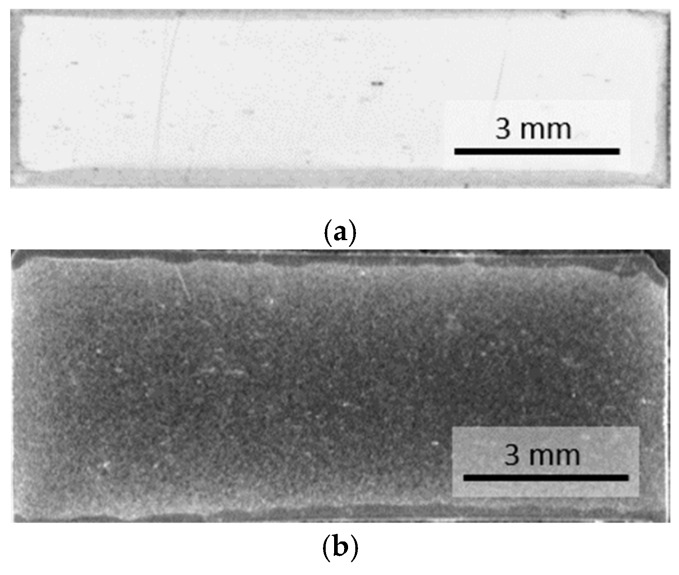
Porosity distribution in: (**a**) a conventional press and furnace sintered compact, and (**b**) a MF-ERS compact (8 kA-400 ms). The clearer areas, more reflecting zones, indicate a lower porosity.

**Figure 10 materials-13-02131-f010:**
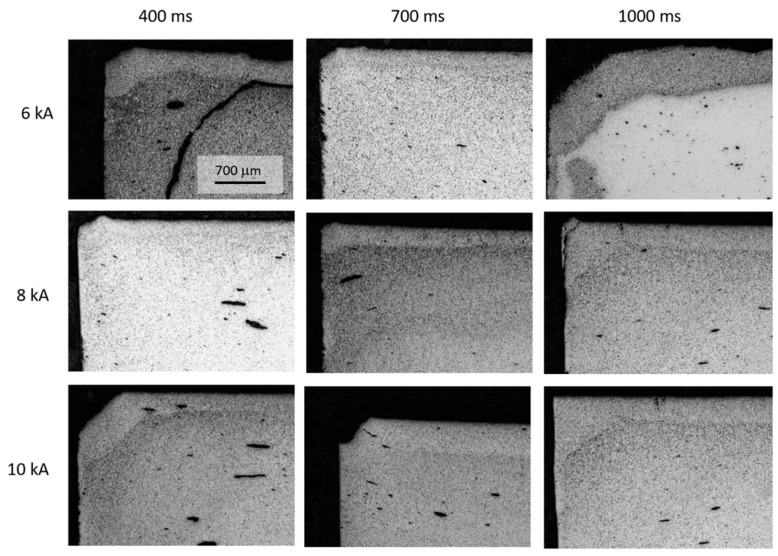
Porosity distribution at a corner of the MF-ERS compacts, as a function of different MF-ERS conditions.

**Figure 11 materials-13-02131-f011:**
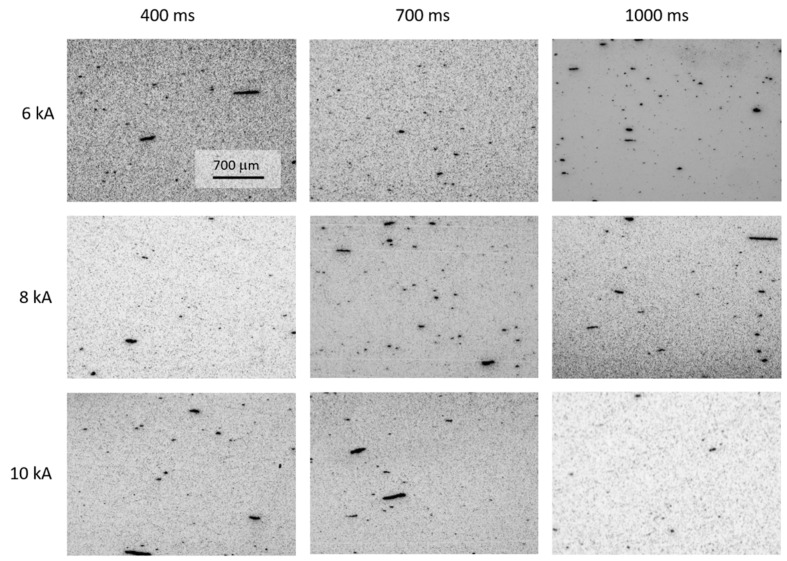
Porosity distribution at the center of MF-ERS compacts, as a function of different MF-ERS conditions.

**Figure 12 materials-13-02131-f012:**
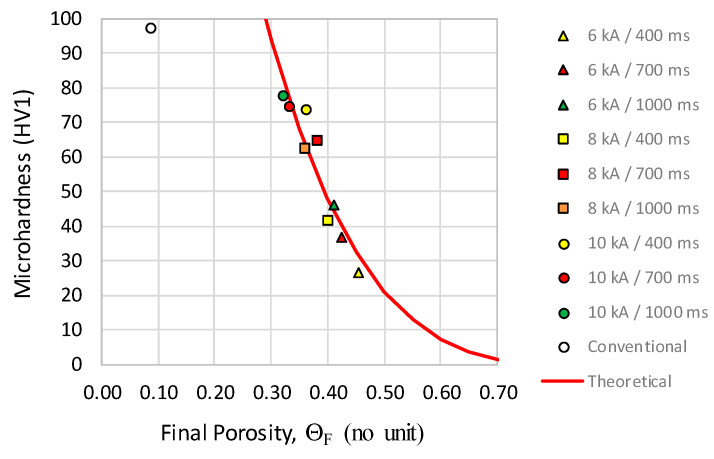
Mean microhardness (HV1) vs. final porosity of the compacts.

**Figure 13 materials-13-02131-f013:**
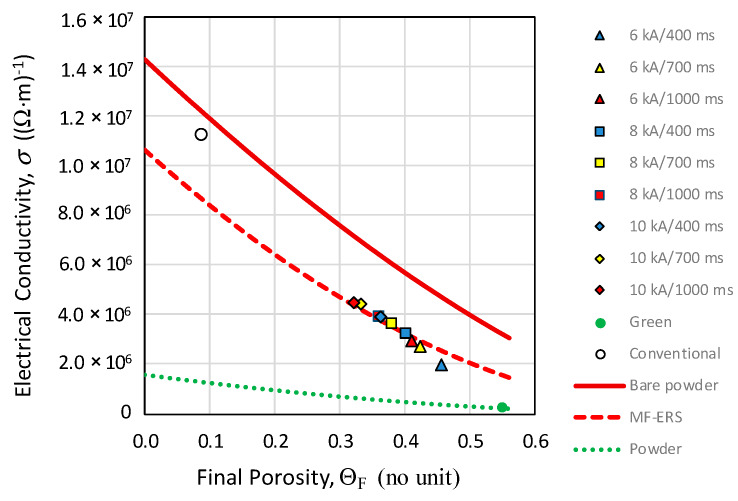
Electrical conductivity vs. final porosity in the MF-ERS compacts, conventional compact and green compact. The theoretical models given by Equation (4) with different values are also represented.

**Table 1 materials-13-02131-t001:** Values of the compact final porosities (Θ_F_), expressed as a fraction, for different processing conditions.

		Heating Time (ms)		
		400	700	1000
**Intensity (kA)**	**6**	0.45	0.42	0.41
	**8**	0.40	0.38	0.36
	**10**	0.36	0.33	0.32

**Table 2 materials-13-02131-t002:** *STE* values (expressed in kJ/g) for the different MF-ERS experiments.

		Heating Time (ms)		
		400	700	1000
**Intensity (kA)**	**6**	0.29	0.42	0.52
	**8**	0.37	0.50	0.61
	**10**	0.39	0.52	0.65

**Table 3 materials-13-02131-t003:** Mean values and standard deviations of microhardness (HV1) for the different MF-ERS compacts.

		Heating Time (ms)		
		400	700	1000
**Intensity (kA)**	**6**	27 ± 9	37 ± 7	46 ± 9
	**8**	42 ± 14	65 ± 5	68 ± 9
	**10**	74 ± 8	75 ± 6	78 ± 6

**Table 4 materials-13-02131-t004:** Values of electrical conductivity *σ*, expressed in (Ω·m)^−1^, for the different MF-ERS compacts.

		Heating Time (ms)		
		400	700	1000
**Intensity (kA)**	**6**	1.97 × 10^6^	2.73 × 10^6^	2.94 × 10^6^
	**8**	3.23 × 10^6^	3.62 × 10^6^	3.88 × 10^6^
	**10**	3.91 × 10^6^	4.44 × 10^6^	4.50 × 10^6^

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
