# Peer review of "Nickel Porous Compacts Obtained by Medium-Frequency Electrical Resistance Sintering"

_materials, 2020, doi:10.3390/ma13092131_

Round 1

Reviewer 1 Report

Materials-782183-Review

1- This team seems to be one of the few teams working on the MF-ERS process for consolidating metallic materials. A large number of works are published by this group on this research topic. It follows that the articles are very similar both in form and sometimes in substance. We often find the same diagrams of equipment, the same types of curves ...
More annoying, we find in this work, no discussion or comparison with other works on this material (Ni) consolidated by other processes. In addition the tendency to self-citation is very important, 9 self-citation on 21 references in total.
2- Certain details must be provided on the consolidation process. An estimate of the consolidation temperature should be given. It must be specified whether consolidation is carried out in the presence of air.
3- It is assumed that there is nickel oxide in the consolidated material. An X-ray diffraction analysis must be carried out to confirm or even quantify this presence. Indeed the properties of the material depend on it.
4- What is the influence of the consolidation process on the size of the grains obtained? For this, a study in electron microscopy on thinned samples is recommended.
5- Minor remarks:
- a repetition on line 121 page 4
- Page 12, lines 316-318: it is indicated that the electrical conductivity decreases when the porosity decreases. The results obtained show the opposite.

In conclusion, this article cannot be accepted in its current version. The authors are invited to revise this version by replying to the comments and suggestions made in this report.

Author Response

Dear reviewer, thank you for detailed review and your interest in improving the content of this manuscript. Please find next our answer to your different comments. We hope the changes included in the manuscript are in accordance with your suggestions.

1- This team seems to be one of the few teams working on the MF-ERS process for consolidating metallic materials. A large number of works are published by this group on this research topic. It follows that the articles are very similar both in form and sometimes in substance. We often find the same diagrams of equipment, the same types of curves ... More annoying, we find in this work, no discussion or comparison with other works on this material (Ni) consolidated by other processes. In addition the tendency to self-citation is very important, 9 self-citation on 21 references in total.

We had not calculated the self-citation ratio. We have reduced their number to the essential. We have also added some references to papers of other groups using similar techniques. The self-citations are now 7/32.

We have added some works that have dealt with electrical consolidation of nickel powder. But the list is short. We don't know any others.

2- Certain details must be provided on the consolidation process. An estimate of the consolidation temperature should be given. It must be specified whether consolidation is carried out in the presence of air.

Yes, MF-ERS experiments are carried out in the air, without a controlled atmosphere. This detail has been added to the manuscript on line 124.

Also, information related to the temperature reached during the process inside the compact has been added. In line 254 the following text has been included:

In order to complete the information, it would be interesting to provide, at least, the values of the mean temperatures inside the compact. Unfortunately, due to the nature of the MF-ERS technology, it is not possible to reliably measure the temperature inside the compact during the process. Only by means of theoretical estimation is it possible to know the achieved temperatures. Using simulators developed in previous works [13-14], it has been possible to establish that the maximum values of the mean temperatures reached, for the analysed conditions, move in the range of 400 to 700 K. These values may seem low, but it must be considered that the local temperature may be much higher in the interparticle contact areas. Unfortunately, there are no reliable theoretical estimates for these local temperatures.

3- It is assumed that there is nickel oxide in the consolidated material. An X-ray diffraction analysis must be carried out to confirm or even quantify this presence. Indeed the properties of the material depend on it.

The presence of very thin oxide films covering the metals is a well-known fact in Powder Metallurgy and the metallic corrosion field. Certainly, readers unfamiliar with this concept may find it surprising that this is taken for granted. In line 341, a short text has been added to justify this (including new references).

The presence of oxide nanometric films on the surface of metallic parts is a well-known fact in P/M and in the Metallic Corrosion field [28-30]. In P/M, these oxide layers covering the powder particles are considered as drawbacks which hinder (and sometimes prevent) the sintering process [31-32].

Unfortunately, we cannot carry out the measurements by XRD that you indicate, because the central facilities of the University are closed, due to the COVID-19 pandemic.

4- What is the influence of the consolidation process on the size of the grains obtained? For this, a study in electron microscopy on thinned samples is recommended.

The issue you raise is of great interest. Currently, we are working on a simulator that will allow to know the evolution of grain size during the MF-ERS process. Unfortunately, the experimental measurements that were required for the calibration of the models were interrupted by the closure of the central facilities of the University, due to the COVID-19 pandemic. Sorry, it is not possible, at now, to obtain the information you are asking.

5- Minor remarks:

- a repetition on line 121 page 4

Done. We have removed the repeated text.

- Page 12, lines 316-318: it is indicated that the electrical conductivity decreases when the porosity decreases. The results obtained show the opposite.

Right, sorry for the mistake. I had ‘resistivity’ in mind instead of ‘conductivity’. The sentence has been corrected.

In conclusion, this article cannot be accepted in its current version. The authors are invited to revise this version by replying to the comments and suggestions made in this report.

A proof-read has been performed in the manuscript, and changes have been included in the new version.

Reviewer 2 Report

The manuscript entitled: 'Nickel Porous Compacts obtained by Medium-Frequency Electrical Resistance Sintering' is written with a clear motive. However, I have the following concerns: 

  • A strong scientific discussion is missing on the factors influencing the density.
  • Fig. 6(a): for the legend on Y-axis - space should be introduced between the text and units... for instance: Pressure[MPa] should be written as Pressure [MPa]
  • Fig. 6 (a) and (b): Legend on Y-axis should accompany by units.
  • Fig. 7: Legend on Y-axis and Fig. 8, 12 and 13: Legend on X-axis (Final Porosity) should accompany by units.
  • Table 1: Final porosity values should accompany by units.

Author Response

Dear reviewer, thank you for detailed review and your interest in improving the content of this manuscript. Please find next our answer to your different comments. We hope the changes included in the manuscript are in accordance with your suggestions.

The manuscript entitled: 'Nickel Porous Compacts obtained by Medium-Frequency Electrical Resistance Sintering' is written with a clear motive. However, I have the following concerns:

  • A strong scientific discussion is missing on the factors influencing the density.

In view of this, the following text has been added in section 3.2, line 297:

It is not easy to predict, for certain processing conditions, the final porosity that the compacts will reach and even less the porosity distribution. The process of the hot densification that takes place during MF-ERS is coupled with the simultaneous release and transfer of heat. In addition to the processing conditions, the number of interconnected factors (powder composition and morphometric aspects, die and punches materials, compact aspect ratio, etc.) is so great that the only way to advance is by means of numerical simulations. Only with these ones, it is possible to predict the temperature and porosity distribution in the system. This will be the work to be carried out, specifically for the tested conditions of this paper, in the near future.

  • Fig. 6(a): for the legend on Y-axis - space should be introduced between the text and units... for instance: Pressure [MPa] should be written as Pressure [MPa]

Done.

  • Fig. 6 (a) and (b): Legend on Y-axis should accompany by units.
  • Fig. 7: Legend on Y-axis and Fig. 8, 12 and 13: Legend on X-axis (Final Porosity) should accompany by units.

We don't quite understand. Porosity is a dimensionless magnitude. You mean we should indicate whether it's expressed ‘as a fraction’ or a ‘percentage? There is no usual symbol for the fraction ‘unit’. In attention to the reader not familiar with this magnitude, we have added its definition, in line 63 of the manuscript.

We have reviewed all the axis legends of all the plots to standardize styles.

  • Table 1: Final porosity values should accompany by units.

Porosity is dimensionless. We have added ‘expressed as a fraction’ in the head.

A proof-read has been performed in the manuscript, and changes have been included in the new version.

Reviewer 3 Report

The paper is well prepared and the research is well organized.  Some minor comments and suggestions:

-The techniques uses high pressure ( a few hundreds MPa) and it has some analogies with cold sintering described in a recent review paper (A review of cold sintering processes, Advances in Applied Ceramics, 1-29, 2020). In cold sintering of metals the inter-particle bonding formation is important in controlling the material properties.   

Electrical conductivity of cold pressed samples (with no discharge) should be added to figure 13 and table 4. This could be a valuable attempt to separate effects of pressure and current/heating.

- The lower hardness of conventional sintered samples compared to ERS should be better discussed. Why ERS has higher  hardness?  is it because of increased dislocation density or because differences in the  chemical composition  (i.e. surface oxides or other reactions occurring when heating in using conventional sintering) ?  

Author Response

Dear reviewer, thank you for detailed review and your interest in improving the content of this manuscript. Please find next our answer to your different comments. We hope the changes included in the manuscript are in accordance with your suggestions.

The paper is well prepared and the research is well organized. Some minor comments and suggestions:

-The techniques uses high pressure (a few hundreds MPa) and it has some analogies with cold sintering described in a recent review paper (A review of cold sintering processes, Advances in Applied Ceramics, 1-29, 2020). In cold sintering of metals the inter-particle bonding formation is important in controlling the material properties.

We didn't know that interesting review paper. We have included it in the reference list.

Electrical conductivity of cold pressed samples (with no discharge) should be added to figure 13 and table 4. This could be a valuable attempt to separate effects of pressure and current/heating.

We have added the value of the electrical conductivity in Figure 13 (with the legend 'green') and in the paragraph that redirects to Table 4, we have added the following text:

(For comparison purposes, the electrical conductivity of conventionally consolidated compact and green compact (without sintering) are 1.12·107 (W·m)-1 and 2.36·105 (W·m)-1, respectively.)

- The lower hardness of conventional sintered samples compared to ERS should be better discussed. Why ERS has higher hardness? is it because of increased dislocation density or because differences in the chemical composition (i.e. surface oxides or other reactions occurring when heating in using conventional sintering)?

We think that the differences are not due to the chemical composition. In the case of conventional sintering, exposure to high temperature is sufficiently long, and the so small initial size of powder, that significant grain growth can be triggered. This would be associated with a considerable decrease in hardness. The MF-ERS process would not show this phenomenon due to its comparatively short duration. It should also not be ruled out that the rapid cooling provided by the MF-ERS technique, in which the metallic electrodes are water-cooled, causes a rapid contraction and high concentration of dislocations and stresses, resulting in higher hardness.

We have added this discussion in the manuscript.

A proof-read has been performed in the manuscript, and changes have been included in the new version.

Round 2

Reviewer 1 Report

Some important questions remain unanswered. Hopefully the authors can work on it after a return to normal life.

Author Response

Some important questions remain unanswered. Hopefully the authors can work on it after a return to normal life.

Thank you for your good wishes.

At the moment, there is no date scheduled for the opening of the central facilities.

Reviewer 2 Report

The authors have addressed some of the comments but are not satisfactory.

  • The unit of porosity should be expressed in (percentage/volume)... if not units are to be expressed then it should be mentioned as (no unit)
  • The discussion is not satisfactory. The authors may consider involving the physical properties of the material to explain the porosity generation along with the entrapped gases/insufficient process parameters.

Author Response

Dear reviewer, thank you again for review and your interest in improving the content of this manuscript. Please find next our answer to your different comments.

The authors have addressed some of the comments but are not satisfactory.

  • The unit of porosity should be expressed in (percentage/volume)... if not units are to be expressed then it should be mentioned as (no unit).

Thanks for the clarification.

Now, in all the graphs in which the porosity appears, it has been added: (no unit).

  • The discussion is not satisfactory. The authors may consider involving the physical properties of the material to explain the porosity generation along with the entrapped gases/insufficient process parameters.

Thank you for clarifying the discussion you required. In relation to this, we have added this extensive explanation:

To reduce the final porosity of the compact, the current intensity and/or the heating time must be increased, and, in principle, the increase of the applied pressure can also help. However, increasing the applied pressure does not always achieve the desired effect. A higher pressure leads to a higher density in the first moments of the heating period. This reduction can be counterproductive as it will lead to a decrease in the electrical resistivity of the powder mass, which will result in less thermal energy being released, and therefore less softening of the material and higher final porosity.

High pressure values can cause another undesirable effect which will also lead to higher final porosity. When the process is carried out with adequate pressure, the centre of the compact, at higher temperature, collapses first. The gas contained in the pores of this zone (or even that which may evolve due to the heating) can be evacuated through the (open) porosity of the peripheral zones, which are kept at a lower temperature. Thus, in the correct operation of the MF-ERS, porosity is progressively removed from the interior towards the outer. The peripheral porous zones will become narrower and the central area denser, increasing uniformity over time. On the other hand, when the pressure value is too high, the porosity of the peripheral zones may close, preventing the subsequent evacuation of the gases inside the compact. The gas that is occluded at a very high temperature can generate enough counter-pressure to prevent the densification of the compact. Therefore, the choice of pressure can be critical, and it cannot be said that increasing it will result in a guaranteed decrease in the final porosity. Therefore, if the aim was to reduce the final porosity of the samples (which was not the case in this paper), intensities and times would have to be increased. Obviously, if conditions are too severe, the compact core could become molten [13].

It can be noticed (Figures 10 and 11) that the pores in the central area generally have a more rounded appearance than those in the peripheral areas. Naturally, this fact becomes more evident as the processing conditions (intensity and heating time) become more severe, as a consequence of the higher temperatures reached.

We hope the changes included in the manuscript are now in accordance with your suggestions.